# A sero-epidemiological analysis of *Coxiella burnetii* infection and its risk factors in livestock from Addis Ababa, Adama, and Modjo abattoirs and pastoral areas of Oromia, Ethiopia

**Ashenafi Milkesa[1], Tesfaye Rufael[2], Getachew Kinfe[2], Redeat Belaineh[2], Abdella Bulbula[2], Donghee Cho[1], Mohammed Naimuddin[1], Teshale Sori[3], Hunduma Dinka[1] ***

1 Department of Applied Biology, School of Applied Natural Science, Adama Science and Technology University, Adama, Ethiopia, **2** Animal Health Institute (AHI), Sebeta, Ethiopia, **3** Department of Clinical Studies, College of Veterinary Medicine and Agriculture, Addis Ababa University, Bishoftu, Ethiopia

* dinkahu@gmail.com

**Data Availability Statement:** All relevant data are within the manuscript.

## Abstract

### Background

*Coxiella burnetii* is causing infections in both humans and animals, resulting in Q fever and Coxiellosis, respectively. Information on the occurrence of *C. burnetii* infection is scarce in Ethiopia. This study estimated the sero-prevalence of *C. burnetii* infection and associated risk factors in four common livestock species from Addis Ababa, Adama, and Modjo abattoirs and pastoral areas of Oromia, Ethiopia.

### Results/principal findings

Sera samples were analyzed for the presence of anti-*C. burnetii* antibodies using an indirect Enzyme Linked Immunosorbent Assay kit. Out of the 4140 serum samples tested, 777 (18.77%; 95% CI: 17.59, 19.99) were found positive for *C. burnetii*. The sero-prevalence estimate was 27.17% at Addis Ababa abattoir, 19.41% at Adama abattoir, 19.13% at Modjo abattoir and 12.1% in animals tested from pastoral areas. Sera analysis at the animal species level showed that cattle exhibited the lowest sero-prevalence estimate (11.83%; 95% CI, 10.27–13.53%), while the highest was observed in camels (28.39%; 95% CI, 25.16–31.80%). The sero-prevalence estimate was 21.34% (95% CI, 18.86–23.99%) in goats and 20.17% (95% CI, 17.49–23.07%) in sheep. The results of multivariable logistic regression analysis showed that species, age, sex of animals and tick infestation were important risk factors for *C. burnetii* infection. The odds of infection were 3.22 times higher in camels and almost twice as high in goats and sheep compared to cattle. Adult animals were infected more likely (OR = 3.23) than young ones. Interestingly, a significant difference was observed in the sero-prevalence of infection between animals that were infested with ticks (OR = 16.32) and those which were tick-free.

**Funding:** The author(s) received no specific funding for this work.

**Competing interests:** The authors have declared that no competing interests exist.

## Conclusion

This study provides valuable insights into the sero-epidemiology of *C. burnetii* infection in four common livestock species at major abattoirs and pastoral areas of Ethiopia. The findings highlight the need for further studies and implementing surveillance and biosecurity measures to prevent the spread of the disease in both humans and livestock to safeguard the economical and public health aspects.

## Author summary

*Coxiella burnetii* causes infections in both humans and animals, resulting in Q fever and Coxiellosis, respectively. This bacterium poses significant public health, veterinary, and economic risks worldwide due to its potential to cause severe illness and low infectious dose. Therefore, it is important to estimate the sero-prevalence of *C. burnetii* infection and associated risk factors in cattle, sheep, goats, and camels, respectively. The serum samples were analyzed for the presence of anti-*C. burnetii* antibodies using an indirect Enzyme Linked Immunosorbent Assay (iELISA) kit. A prevalence estimate of 18.77% was observed for *C. burnetii*. The sero-prevalence estimate at abbatoirs and pastoral areas was found in the order of Addis Ababa (27.17%)>Adama (19.41%)>Modjo (19.13)>pastoral areas (12.1%). The risk factors were identified as animal species, age, sex, and tick infestation. Camels were more vulnerable than sheep, goats and cattle. The adults were infected more than young. The ticks may be a potential vector for the transmission of the pathogen.

## Introduction

*Coxiella burnetii* is a bacterium that causes Q fever in humans [1,2] and coxiellosis in animals [3,4]. Transmission is primarily through inhalation of dry aerosol particles from infected animals although tick associated transmission has also been reported in animals [5]. Coxiellosis is known to cause reproductive losses in livestock and debilitating illness in humans [2]. For instance, the outbreak that occurred in the Netherlands between 2007 and 2010 resulted in the culling of over 58,000 (~21%) livestock and illness in more than 4,000 people [6].

Despite its significant impact on both animal and human health, *C. burnetii* is often overlooked in developing countries including Ethiopia. Since its identification in southern Moroccan ticks in 1947, only limited investigations have been conducted in Africa, primarily due to challenges in the diagnosis of infection, insufficient resources and/or expertise, and prioritization of other infectious diseases [7,8]. The seroprevalence of *C. burnetii* infection has been noted to vary considerably in African countries; higher seroprevalence observed in countries with greater pastoral land use, such as Chad, Egypt, Tanzania, Niger, and Kenya [8–10].

The first detection of *C. burnetii* in Ethiopia dated back to 1966 [11]. Subsequently, it was reported in 6.5% of abattoir workers in Addis Ababa in 1990 [12] and in cases of infectious endocarditis [7]. Gumi and his colleagues also reported evidence of infection in ruminants and camels in pastoral areas, with seroprevalence of 31.6% in cattle, 90% in camels, and 54.2% in goats [13]. A single study pointed out the association of *C. burnetii* infection with abortion in animals [7].

Many countries which have conducted serological surveys had determine the prevalence of *C. burnetii* in domestic ruminants using various serological techniques such as complement fixation test, immunofluorescence assays, and enzyme-linked immunosorbent assays. For example, serological investigations on coxiellosis were conducted in Europe, Latin America and several African countries. The presence of *C. burnetii* in several animal species linked to its zoonotic importance has been justified. The reservoirs of *C. burnetii* have been largely investigated in many countries, especially in Europe. The occurrence of *C. burnetii* in wildlife has also been evident [2,3,14]. In Ethiopia few serological studies were conducted in few locations focusing on a small number of animals and using various sampling procedures [12,13]. Therefore, this study aimed to estimate the seropevalence of coxiellosis in four common livestock species originated from various areas of Ethiopia and determine the associated risk factor for *C. burnetii* infection.

## Materials and methods

### Ethics statement

Sampling of animals followed the approved experimental procedures and standards established by Animal Research Scientific and Ethics Review Committee (ARSERC) at the Animal Health Institute (AHI), the Institutional Animal Care and Use Committee (IACUC), and Adama Science and Technology University (ASTU) with certificate Ref. No. RECSoANS/BIO/11/2022.

### Description of the study area

The research was carried out with animals destined for slaughter at Addis Ababa, Adama, and Modjo abattoirs, as well as from the Borana pastoral area of Oromia, which are depicted in Fig 1. The Borana pastoral area is located in the southern part of Ethiopia's Oromia Regional State. The capital of Borana Zone, Yabello, is situated 575 kilometers south of Addis Ababa. The Borana Zone has thirteen districts and share borders with Kenya in the southern part at Moyale, Miyo, Dirre, and Teltelle districts. For this study, three districts were chosen randomly, that included Yabelo, Dubuluk, and Mega. The Borana Zone has a semi-arid to arid climate and is geographically located between 4° to 6° N latitude and 36° to 42° E longitude, featured with isolated mountains and valleys. The altitude of the Borana Zone ranges from 1,000 to 1,700 meters above sea level, and the mean annual rainfall in the area ranges from 250 to 700 mm. The average annual temperature in the area ranges from 19 to over 25°C. The Borana people rely mainly on extensive pastoralism, or nomadic herding, as their primary means of livelihood. The main livestock species raised in the region comprises cattle, goats, sheep, and camels. The Borana Zone Pastoral Development Office [15] reports that the zone has livestock population of 1,416,180 cattle, 1,262,782 goats, 776,870 sheep, 237,205 camels, 306,057 poultry, 102,767 donkeys, 1,841 horses, and 4,433 mules. As of 2015, the human population was reported to be 1,283,925 [16].

The Addis Ababa abattoir Enterprise in Addis Ababa is situated in the central highlands of Ethiopia at an altitude of 2500 meters above sea level. The average annual temperature and rainfall in the city are 21°C and 1800 mm, respectively. During the rainy season, the relative humidity ranges from 70 to 80%, while during the dry season, it ranges from 40 to 50%. According to the CSA [17] of Addis Ababa City Administration Agricultural Report, an average of 183,000 cattle, 42,200 sheep, 4,700 goats, and 830 pigs are slaughtered at the abattoir each year. Generally, the cattle, sheep, and goats that are slaughtered in the abattoir come from different regions and agro-ecological zones across the country.

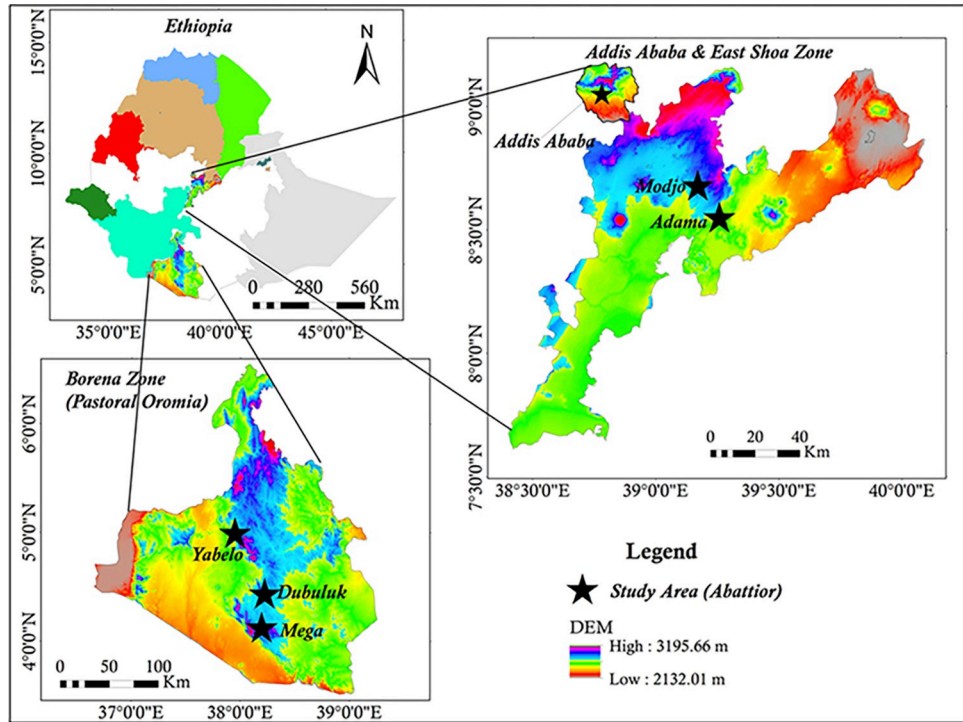

**Fig 1. Map of Ethiopia depicting the locations of the study areas ([http://www.usgs.gov](http://www.usgs.gov)).**

Adama town is a significant town located in the East Shoa Zone of Ethiopia's Oromia region, situated at 8.54˚N and 39.27˚E, with an elevation of 1712 meters and approximately 99 kilometers southeast of Addis Ababa. It is one of the largest metropolitan cities in Ethiopia, with an estimated population of 450,000 humans and many resorts that make it suitable for conferences and tourism. Adama abattoir has the capacity to slaughter approximately 150 cattle and 500 sheep and goats per day [17].

Modjo is the administrative center of Lume district, which is located in the East Shoa Zone of Ethiopia's Oromia Region. It is located 66 kilometers southeast of Addis Ababa, at latitude 8˚35'N and longitude 39˚7'E, at an elevation of 1790 meters above sea level. The area receives rainfall twice a year, during the long and short rainy seasons. The rainy season lasts from June to September. The average annual rainfall, temperature, and mean relative humidity are 776 mm, 19.4˚C, and 59.9%, respectively [17]. The livestock used in this study were all local breeds and mostly came from different agro-ecological zones, including Arsi, Borana, Jimma, Somali, and South Wello, respectively. These animals were raised under extensive production systems, either as part of a mixed crop-livestock production system or within a pastoral system of production. They were acquired from various local markets and transported to the abattoirs, where they were given food, water, and rest for a period ranging from 24 to 72 hours before being slaughtered [18].

## Study population and design

A cross-sectional study was conducted from January 2021 to May, 2022. The study population comprise cattle, sheep, goat and camel population raised in the catchment areas of the abattoirs (Addis Ababa, Adama, and Modjo) and Borana pastoral area. Animals of both sexes aged more than 6 months and clinically healthy animals from selected abattoirs and three selected

pastoral areas, under the observation of veterinary workers, were the study population. The age of the study animals was estimated as previously described by Schelling and his colleagues [19].

After having a list of pastoralist households in Borana, clustered households and their livestock were chosen randomly for sampling from households and their herds. In the Borana pastoral area, the majority of animals included in the study were females as they were used for milk production purpose. In contrast, the majority of animals sampled at abattoirs were male livestock. All the study animals were not vaccinated against *C. burnetii*.

## Sample size determination

To determine the minimum number of animals needed for this study, a formula developed by Thrusfield [20] was employed. In this formula, n represents the required sample size, $P_{exp}$ is the expected prevalence, and d is the absolute precision as indicated in Eq (1).

$$n = \frac{(1.96)^2 \text{ X } P_{eep}(1 - P_{exp})}{d^2} \quad (1)$$

where, *n* is the total sample size, $P_{exp}$ is expected prevalence (50%), *d* is absolute precision (0.05) at 95% CI.

$$n = \frac{(1.96)^2 \text{ X } 0.5(1 - 0.5)}{(0.5)^2} = 384$$

The study required a total of 1536 animals, with 384 animals needed for each of the four animal species. In order to increase precision and decrease the standard error, a total of 4,140 animals (1,564 cattle, 736 camels, 1,012 goats, and 828 sheep) were utilized in this study. Of these, 1,012 samples were obtained from Addis Ababa abattoir, 1,511 from Modjo abattoir, and 237 from Adama abattoir, while 1,380 animals were sampled from pastoral areas.

## Blood sample collection and serum separation

To make the sample collection process easier, it was preferred to collect the samples early in the morning before the animals were taken out for grazing or slaughter, depending on whether they were from pastoral areas or abattoirs. During the blood sample collection process, the animals were held in a crush to limit their mobility. Aseptic blood samples, approximately 6–8 mL, were collected from each animal's jugular vein by using sterile plain vacutainer tubes and needles. Each sample was given a code to identify the specific animal, and protected from direct sunlight.

The samples were placed in a slant position overnight at room temperature to allow clotting. The next morning tubes containing samples were centrifuged at 2000× g for 10 min at room temperature by taking them to the nearby veterinary clinics, and serums were separated from the blood and stored in 1.8 mL disposable screw-capped cryovials (Cryo.STM; Greiner Bio-one, GmbH Frickenhausen, Germany). Then the samples were transported to Animal Health Institute (AHI) laboratory in Sebeta, Ethiopia, at a temperature of 4˚C using a portable refrigerator that was plugged into a car and subsequently stored at -20˚C until they were analysed.

Information about the individual animal such as its area, sex, age, species, and tick infestation, was recorded during the sample collection process by using sample information collection sheet prepared for this purpose.

## Serological examination

The commercial indirect enzyme-linked immunosorbent assay (iELISA) kit (Switzerland AG, CH3097, Liebefeld-Bern, Switzerland) was used to detect antibodies to phase I antigens of *C. burnetii*. The antigen coated wells of the ELISA plates were incubated with the diluted serum samples according to the manufacturer's instructions. The negative and positive controls were diluted at 1:400 using wash solution and dispensed into duplicate wells, while the serum samples were added to the remaining wells. The plate was covered and gently shaken and incubated for 60 min at 37°C in a humid chamber. After incubation, the wells were emptied and washed three times with 300 μL of wash solution, and then 100 μL of secondary antibody conjugate was added and incubated as described above. The plate contents were discarded, and the plate was washed three times using 300 μL of wash solution. Subsequently, 100 μL of substrate was added, and the plate was incubated at room temperature for 15 min. Finally, 100 μL of stop solution was added, and the absorbance was read at 450 nm.

The Agilent BioTek ELx800 absorbance microplate Reader was employed to measure the absorbance values as Optical density (OD). The S/P-ratio, which represents the percentage of the ratio between the sample OD and the positive control OD, was calculated as follows: Samples with an S/P percent ≥40% were considered seropositive for *C. burnetii*, while those with values between 30–40% were considered dubious, and those ≤30% were considered negative. Based on the sensitivity (99%) and specificity (98%) of the commercial ELISA kit we used, the true prevalence of *C. brnetii* infection was also calculated using the formula previously described by Farrokh et al. [21].

## Geo-referencing

The animals that were tested were distributed across the areas/districts from where they originated. During the study period, they were sampled and analyzed using thematic density maps created using QGIS 10.0 [22]. The thematic density maps were used to show the distribution of the animal samples collected during the study period, as well as the distribution of the animals that tested positive for *C. burnetii*. Two thematic maps were created to show the distribution of the samples: one indicating the total number of samples, and another indicating the number of serologically-positive samples identified by using iELISA.

## Data management and analyses

The STATA software version 16.0 (Stata Corp, 4905 Lake way Drive, College Station, Texas 77845 USA) was used to analyse the data and determine the effects of risk factors on the prevalence of *C. burnetii* infection. Test statistics such as chi-squared test was to analyse the effect of individual factors on the prevalence of *C. burnetii* infection. Multivariable logistic regression was employed to compute the associations between the risk factors and prevalence. The variables were manually fitted to the model to check for confounding. To evaluate the fitness of the logistic regression model, the Hosmer-Lemeshow and Pearson methods were used. A P value of ≤0.05 and a 95% confidence interval were considered to be significant in this study.

## Results

### Seroprevalence of *C. burnetii* infection

The present study analysed a total of 4140 sera samples to estimate the prevalence of antibodies against *C. burnetii*. The overall seroprevalence of *C. burnetii* infection was 18.77% (95% CI, 17.55–19.99%) (Table 1). Hence, based on the sensitivity (99%) and specificity (98%) of the commercial ELISA kit we used, the calculated true prevalence of *C. brnetii* infection was

**Table 1. The seroprevalence of *Coxiella burnetii* infection determined in livestock sampled from diverse geographic localities.**

| Variable | Category | No. tested | No. positive | Prevalence (%) | $X^2$ | 95% CI | |
|---|---|---|---|---|---|---|---|
| | | | | | | Lower | Upper |
| Area | Addis Ababa Abattoir | 1012 | 275 | 27.17 | 87.32 | 24.45 | 30.03 |
| | Modjo Abattoir | 1511 | 289 | 19.13 | | 17.17 | 21.2 |
| | Adama Abattoir | 237 | 46 | 19.41 | | 14.57 | 25.03 |
| | Borana pastoral areas | 1380 | 167 | 12.1 | | 10.43 | 13.94 |
| Species | Cattle | 1564 | 185 | 11.83 | 99.63 | 10.27 | 13.53 |
| | Camel | 736 | 209 | 28.39 | | 25.16 | 31.8 |
| | Goat | 1012 | 216 | 21.34 | | 18.86 | 23.99 |
| | Sheep | 828 | 167 | 20.17 | | 17.49 | 23.07 |
| Sex | Female | 683 | 182 | 26.65 | 33.31 | 23.37 | 30.13 |
| | Male | 3457 | 595 | 17.21 | | 15.97 | 18.51 |
| Age | Young | 1850 | 204 | 11.03 | 131.46 | 9.64 | 12.54 |
| | Adult | 2290 | 573 | 25.02 | | 23.26 | 26.85 |
| Tick infestation | Yes | 142 | 115 | 80.99 | 378.48 | 73.55 | 87.08 |
| | No | 3998 | 621 | 7.05 | | 6.52 | 7.59 |
| **Total** | | **4140** | **777** | **18.77** | | **17.59** | **19.99** |

17.29%. The lowest seroprevalence was recorded in animals from pastoral areas (12.1%; 95% CI, 10.43–13.94%) while the highest prevalence observed in animals from Addis Ababa abattoir (27.17%; 95% CI, 24.45–30.03%). Sera analysis at the animal species level showed that cattle exhibited the lowest seroprevalence (11.83%; 95% CI, 10.27–13.53%), while the highest was observed in camels (28.39%; 95% CI, 25.16–31.80%). The seroprevalence was 21.34% (95% CI, 18.86–23.99%) in goats and 20.17% (95% CI, 17.49–23.07%) in sheep.

Female animals were observed to have a higher seroprevalence (26.65%; 95% CI, 23.37 30.13%) than male animals (17.21%; 95% CI, 15.97–18.51%). The prevalence was also higher in adult animals (25.02%; 95% CI, 23.26–26.85%) than their younger counterparts (11.03%; 95% CI, 9.64–12.54%). Higher seroprevalence of *C. burnetii* infection was also observed in animals infested with ticks (carry ticks on several body parts) (80.99%; 95% CI, 73.55–87.08%) than those animals which were not infested by ticks at all (7.05%; 95% CI, 6.52–7.59%).

## Geo-referencing the seroprevalence of *C. burnetii* from Abattoirs

Abattoirs serve as a valuable source of information on the prevalence of *C. burnetii* infection in different regions of the country, as the animals that are slaughtered there come from various parts of the country. In order to provide insight into the distribution of seroprevalence results across the country, the data obtained from the animals slaughtered at different Abattoirs were geo-referenced by thematic density maps created by using QGIS 10.0 and presented in Fig 2. This spatial information is critical in identifying areas with a higher risk of *C. burnetii* infection, which may serve to inform targeted interventions to mitigate the spread of the bacterium and reduce the risk of transmission to humans. Consequently, this data may play a pivotal role in enhancing public health outcomes and reducing the burden of zoonotic diseases.

In the context of Addis Ababa Abattoir (AAA), animals (sample size, 1012) were predominantly sourced from three distinct regions, which are Amhara (East and West Gojam; North and South Gondar), Oromia (Bale and Borana), and Southern Nations Nationalities and People Region (SNNPR) (Wolayita). These regions were identified and are graphically represented in Fig 2A. For Adama Abattoir (AA), animals (sample size, 237) were procured from a diverse range of regions from Somali, Oromia, Afar, Babile, and Jinka, as illustrated in Fig 2B. These

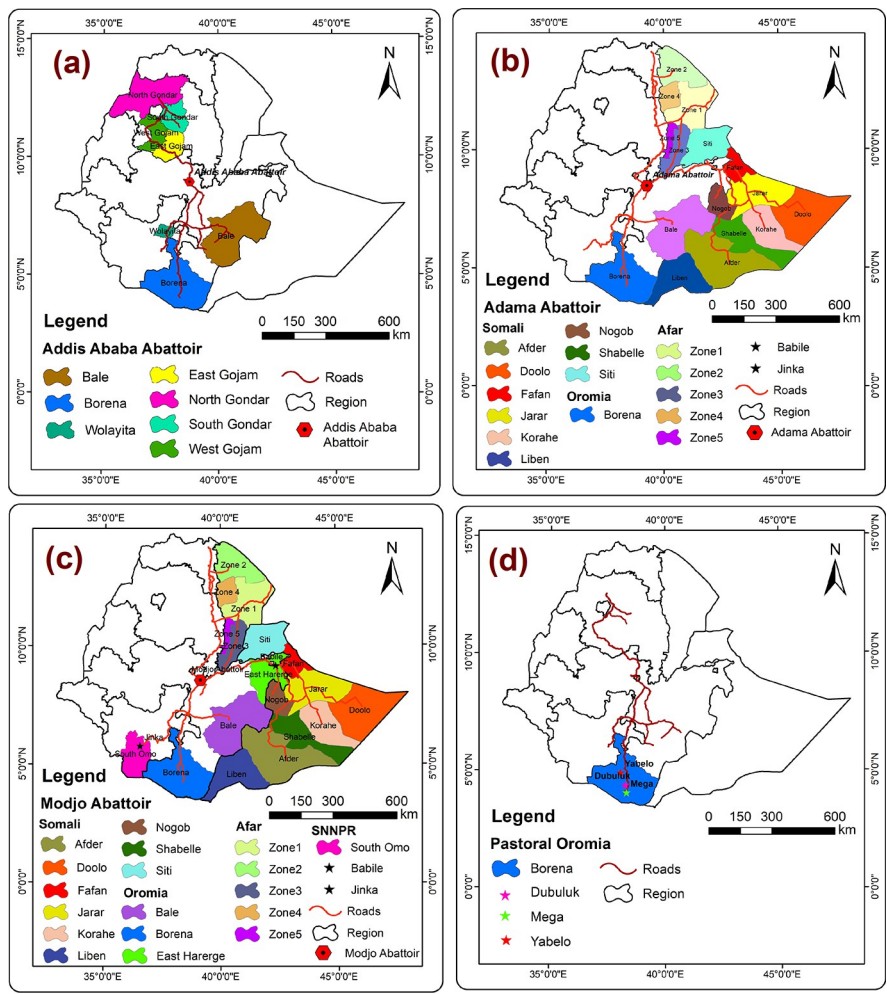

**Fig 2. The serum samples collected from animals were systematically categorized according to their respective sampling areas and sources.** The four sampling areas were as follows: (a) Addis Ababa Abattoir (AAA), which is located in the capital city of Ethiopia, (b) Adama Abattoir (AA), which is situated in the city of Adama, (c) Modjo Abattoir (MA), located in the town of Modjo, and (d) Borana Pastoral Oromia (PO), which is a pastoral region in Ethiopia. https://www.ethiogis-mapserver.org/.

regions were identified and mapped to provide a comprehensive understanding of the geographical origins of the animals. For Modjo Abattoir (MA), animals (sample size, 1511) were primarily sourced from regions similar to Adama Abattoir (AA), with an additional procurement from Bale, East Harerge in Oromia, and the Southern Nations Nationalities and People Region (SNNPR), as depicted in Fig 2C. In the Borana Pastoral region of Oromia, animals (sample size, 1038) were sourced from Dubuluk, Mega, and Yabelo, as depicted in Fig 2D. This geographical information is crucial in determining the epidemiology of infectious diseases that may affect the animals and subsequently pose a risk to human health. By utilizing this data, targeted interventions may be designed to mitigate the risk of zoonotic disease transmission, thereby improving public health outcomes.

The geographical distribution of *C. burnetii* infection was analysed and is presented in Fig 3. The prevalence was found to be higher in animals at the Addis Ababa Abattoirs than those at the other two Abattoirs and pastoral region. The tested sample size and prevalence in case of Addis Ababa Abattoir were 1012 and 27.12%, Adama Abattoir were 237 and 19.41%,

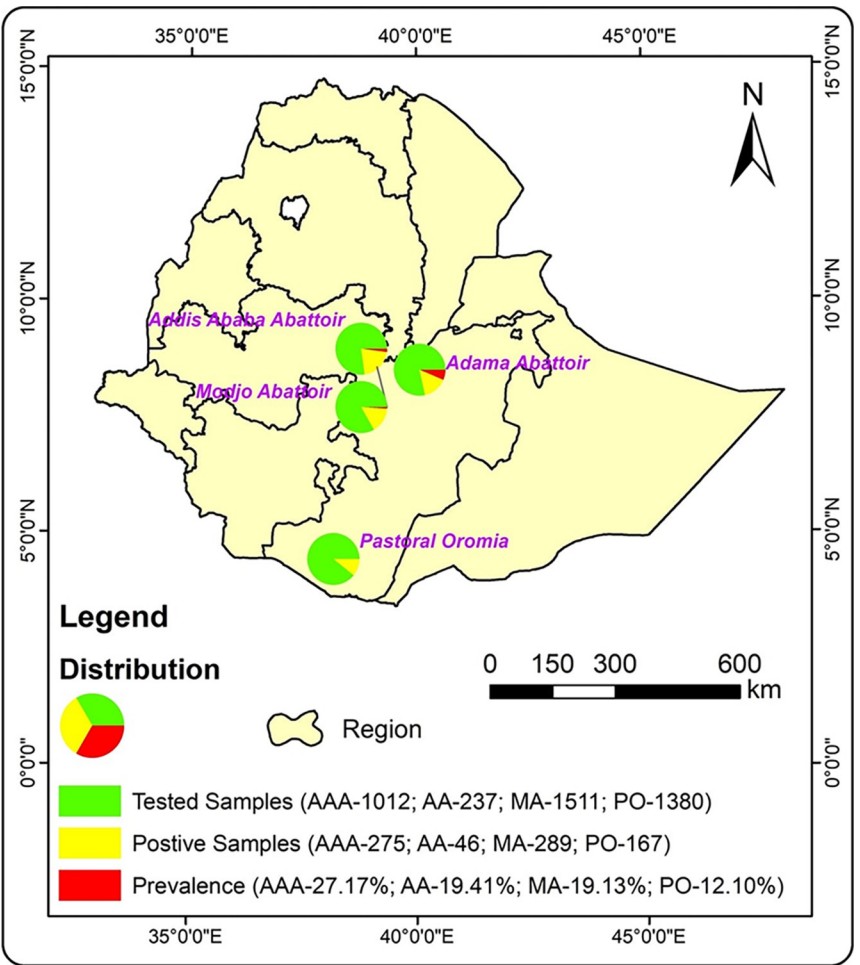

**Fig 3. The seropositivity of animals infected with *Coxiella burnetii* was evaluated using the Indirect Enzyme-linked Immunosorbent assay (iELISA) method in four distinct sampling regions: Addis Ababa Abattoir (AAA), Adama Abattoir (AA), Modjo Abattoir (MA), and Pastoral Oromia (PO).** The study presented the distribution of samples tested, positive, and prevalence percentage, which were represented using different colors. https://open.africa/dataset/ethiopia-shapefiles.

Modjo Abattoir were 1511 and 19.13% and Borana Pastoral Oromia were 1380 and 12.1%, respectively (Fig 3 and Table 1).

### Analysis of risk factors of *C. burnetii* infection

Statistically significant association ($p < 0.001$) was found between animal species and the sero-prevalence of *C. burnetii* (Table 2). The odds of infection in camels was 3.22 (95% CI, 2.57–4.05) times that of cattle. Sheep and goats were twice more likely to be infected with *C. burnetii* than cattle (for sheep, OR = 2.15; 95% CI, 1.63–2.82%; for Goats, OR = 1.99; 95% CI, 1.53–2.6%). The adult animals had higher odd of infection than younger animals (OR = 3.23; 95% CI, 2.63–3.96%; and $p < 0.001$). Those animals infested with ticks were more likely to be sero-positive to *C. burnetii* than animals which were free of ticks. The odds of infection in animals carrying one or more ticks was 16.32 times (95% CI, 10.17–26.18%) the odds of infection in animals that were not infested by ticks. Additionally, a higher prevalence was observed in the female animals compared with male animals, however, the difference in prevalence between

**Table 2. Results of a multivariable logistic regression analysis examining the association between various variables for the seroprevalence of *Coxiella burnetii* infection in livestock.**

| Variables | Category | OR | SE | Z | p-value | 95% CI | |
|---|---|---|---|---|---|---|---|
| | | | | | | Lower | Upper |
| Species | Cattle | Ref | | | | | |
| | Camel | 3.22 | 0.374 | 10.09 | 0.000 | 2.56835 | 4.04817 |
| | Goat | 1.99 | 0.27 | 5.11 | 0.000 | 1.5315 | 2.60322 |
| | Sheep | 2.15 | 0.299 | 5.5 | 0.000 | 1.63525 | 2.82048 |
| Sex | Female | Ref | | | | | |
| | Male | 0.78 | 0.108 | -1.81 | 0.071 | 0.59126 | 1.02174 |
| Age | Young | Ref | | | | | |
| | Adult | 3.23 | 0.337 | 11.24 | 0.000 | 2.63228 | 3.96245 |
| Tick infestation | Yes | 16.32 | 3.935 | 11.58 | 0.000 | 10.1726 | 26.1781 |
| | No | Ref | | | | | |
| | Constant | 0.07 | 0.012 | -15 | 0.000 | 0.04674 | 0.09504 |

OR, Odds Ratio; CI, Confidence Interval; Ref., Reference; SE, Standard Error; Z, z-score.

the two sexes was not statistically significant (OR = 0.78; 95% CI, 0.59–1.02). The results of the multivariable logistic regression analysis are presented in Table 2.

## Discussion

Currently, there is insufficient information on the epidemiology of *C. burnetii* in livestock species in Ethiopia. The present study estimated the seroprevalence of coxiellosis and determined its risk factors in four livestock species originating from major abattoirs and pastoral areas of Ethiopia. Notably, the abattoir study was conducted 56 years after the first evidence of the disease was reported in goats and sheep slaughtered at the Addis Ababa abattoir [11]. The findings of present study revealed that coxiellosis is prevalent in all the livestock species tested across a wider geographical area, indicating a significant public health risk.

The current study reports an overall seroprevalence of 18.77% (95% CI, 17.59–19.99%) with varying levels from the tested livestock species of various locations in Ethiopia. Our finding is in a close similarity with the 20% prevalence reported from south-western Ethiopia using similar serological methods [23]. However, it is lower than those reported by Gumi et al. [13] and Tesfaye et al. [24] who estimated a prevalence of 31.6% and 28.46%, respectively, in other parts of Ethiopia. Some studies from other countries by Cekani et al. [14] in Albania (9.1%), Mwololo et al. [25] in Kenya (12.80%), and Gummow et al. [26] in South Africa (7.78%), also reported a lower seroprevalence than our observation. The variation in seroprevalence observed between our study and the previous ones could be attributed to differences in sample size, sampling methodology, and diagnostic tests utilized. Although some of the previous studies used similar serological tests, they had smaller sample sizes [25,27]. Other possible reason for the variation could be differences in geographical locations and management practices [28]. It is, therefore, crucial to consider these factors when interpreting the results of serological studies on the prevalence of *C. burnetii* infection in animal populations.

In this study significantly higher prevalence of *C. burnetii* infection was observed in camels' goats, and sheep than cattle. In consent to our finding, previous studies have demonstrated significantly higher seroprevalence of *C. burnetii* infection in camels than in other livestock species in Kenya [29], the Sahel region [19], and Ethiopia [13]. This variation in seroprevalence among different animal species could be attributed to differences in susceptibility to *C. burnetii*

infection, can vary based on several factors, including the immune system, age, sex, and management practices [7]. Previous studies have revealed variations in seroprevalence among different animal species [30], where transmission dynamics and environmental factors differ. The higher seroprevalence in camels may be attributed to several factors. Camels are known to have longer lifespans (~40 yrs) than other livestock species [31], and therefore, they may have greater exposure to *C. burnetii*. The prevalence recorded in sheep and goats is consistent with the results of previous studies conducted in Ethiopia and elsewhere in the world [3,13,32–34]. The lower seroprevalence in cattle observed in the present study is also in agreement with the reports of previous studies [35,36]. Similarly, lower prevalence was reported from few African countries such as Togo [37] and Chad [19]. However, higher seroprevalence is reported from other countries by Adesiyun et al. [38] who reported prevalence of 59.8% in dairy cows from Nigeria, Kelly et al. [39] who recorded 39% in cattle and 10% in goats from Zimbabwe and Hussien et al. [40] who reported prevalence of 64.5% in camels and 29.9% in cattle from Sudan.

In our study, we observed a significantly higher seroprevalence of *C. burnetii* infection in adult animals (OR = 3.23; P < 0.001) than in younger animals. This finding is in agreement with the results of several previous studies [25,13,35]. The higher seroprevalence of *C. burnetii* infection in older animals may be attributed to the cumulative exposure to the pathogen over time. Older animals have a higher likelihood of being exposed to the pathogen through contact with infected animals or the environment, and repeated exposure may increase the likelihood of infection and development of antibodies. This is supported by previous studies demonstrating a higher prevalence of *C. burnetii* infection in animals with a longer exposure history [3]. It could also be attributed to the pathogen's ability to infect the reproductive tissues of animals, leading to a higher dissemination and impact in older animals, which have a longer reproductive history [41].

Our study demonstrated a significant difference in the seroprevalence of *C. burnetii* infection between animals infested with ticks and those which were tick free. This observation is consistent with the reports of previous studies suggesting the possible role of ticks in the transmission and spread of the bacterium [41,5]. It has been elucidated that ticks infesting both wild and domestic animals can transmit *C. burnetii* to both animals and humans [41,42] although only two experimental studies evaluated the vector competence of ticks for *C. burnetii*.

Our study revealed a higher seroprevalence of *C. burnetii* infection in animals slaughtered at Addis Ababa abattoir. This could be attributed to the fact that Addis Ababa abattoir receives animals from all corners of the country mainly from the three largest regions where over 70% of the national livestock are located. This observation implies that Addis Ababa abattoir is a high-risk area for the transmission of *C. burnetii* to abattoir workers and the community. Implementation of biosecurity measures such as personal protective equipment is needed. Moreover, there should be screening of animals before slaughter for zoonotic diseases including *C. burnetii* infection in order to protect the public health.

The geo-referencing maps done in the present study can provide useful insights into the spatial distribution of the phenomenon being studied, in this case, the prevalence of coxiellosis in indigenous Ethiopian cattle, sheep, goats, and camels. The maps can also help researchers to identify areas where the infection is more prevalent, which can assist with disease control and prevention efforts.

Our study has some limitations. We used an indirect ELISA to test the presence of antibodies against *C. burnetti* and could not distinguish between historical exposure and active infections. Additionally, the tests used had less than 100% sensitivity and specificity which could pose a risk of misclassification. Our study did not include environmental factors as covariates

when investigating factors associated with coxiellosis antibody seropositivity, which may have accounted for some of the observed variations across different study sites. Furthermore, serological cross-reactions might have occurred between *C. burnetii* and other bacteria.

In conclusion, the current study reported the seroprevalence of *C. burnetii* in four common livestock species which is significantly higher in camels, sheep and goats than in cattle. It was also significantly higher in females than males, in adults than younger animals and in tick infested animals than tick free animals. Our findings highlight the importance of *C. burnetii* in livestock, which can pose serious public health risks. Further investigation into the economic impacts of *C. burnetii* and establishment of molecular diagnosis and surveillance of the disease both in animals and human is warranted for the design of effective control strategies.

## Acknowledgments

We would like to express our gratitude to Addis Ababa, Adama, and Modjo Abattoirs, as well as the Borana Zone Administration office for their collaboration and support during the data collection process from livestock in various districts. We extend our appreciation to Defense Threat Reduction Agency (DTRA), Adama Science and Technology University (ASTU), and the Animal Health Institute (AHI) for technical and chemicals supports received. In addition, we would like to acknowledge the contributions of the following scientists from the National Animal Health Institute (AHI): Solomon Gebre, Asamenew Tesfaye, Fasil Aklilu, Hagos Asgedom, Getachew Tuli, Matios Lakew, and Delesa Damena. We are also grateful to the scientists at Addis Ababa University (AAU), Naval Medical Research Center (NMRC), Viral and Rickettsial Diseases Department, Silver Spring, including Christina M. Farris and MAJ Amanda Christy.

## Author Contributions

**Conceptualization:** Ashenafi Milkesa, Tesfaye Rufael, Redeat Belaineh, Donghee Cho, Hunduma Dinka.

**Data curation:** Ashenafi Milkesa, Mohammed Naimuddin, Teshale Sori, Hunduma Dinka.

**Formal analysis:** Ashenafi Milkesa, Teshale Sori.

**Funding acquisition:** Tesfaye Rufael, Redeat Belaineh, Hunduma Dinka.

**Investigation:** Ashenafi Milkesa, Getachew Kinfe, Abdella Bulbula.

**Methodology:** Ashenafi Milkesa, Getachew Kinfe, Abdella Bulbula, Hunduma Dinka.

**Project administration:** Tesfaye Rufael, Redeat Belaineh, Hunduma Dinka.

**Resources:** Ashenafi Milkesa, Tesfaye Rufael, Redeat Belaineh.

**Software:** Ashenafi Milkesa, Teshale Sori, Hunduma Dinka.

**Supervision:** Tesfaye Rufael, Redeat Belaineh, Donghee Cho, Hunduma Dinka.

**Validation:** Ashenafi Milkesa, Getachew Kinfe, Redeat Belaineh, Hunduma Dinka.

**Visualization:** Ashenafi Milkesa, Redeat Belaineh, Teshale Sori, Hunduma Dinka.

**Writing – original draft:** Ashenafi Milkesa, Tesfaye Rufael, Redeat Belaineh, Teshale Sori, Hunduma Dinka.

**Writing – review & editing:** Ashenafi Milkesa, Donghee Cho, Mohammed Naimuddin, Teshale Sori, Hunduma Dinka.

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
