## [Decision Letter · Decision Letter 0]

25 Oct 2023

Dear Mr. Milkesa,

Thank you very much for submitting your manuscript "A comprehensive analysis of Coxiella burnetii infection (Q fever) and its risk factors in livestock at Addis Ababa, Adama, and Modjo abattoirs, and Pastoral areas of Oromia, Ethiopia: a seroepidemiological study" for consideration at PLOS Neglected Tropical Diseases. As with all papers reviewed by the journal, your manuscript was reviewed by members of the editorial board and by several independent reviewers. In light of the reviews (below this email), we would like to invite the resubmission of a significantly-revised version that takes into account the reviewers' comments. 

We cannot make any decision about publication until we have seen the revised manuscript and your response to the reviewers' comments. Your revised manuscript is also likely to be sent to reviewers for further evaluation.

Sincerely,

Wen-Ping Guo

Academic Editor

Elsio Wunder Jr

Section Editor

Reviewer's Responses to Questions

**Key Review Criteria Required for Acceptance?**

**Methods**

-Are the objectives of the study clearly articulated with a clear testable hypothesis stated?

-Is the study design appropriate to address the stated objectives?

-Is the population clearly described and appropriate for the hypothesis being tested?

-Is the sample size sufficient to ensure adequate power to address the hypothesis being tested?

-Were correct statistical analysis used to support conclusions?

-Are there concerns about ethical or regulatory requirements being met?

Reviewer #1: The study used a preliminary method for the diagnosis of Q fever. There is no molecular investigation. 

Samples were collected from abattoirs and three selected pastoral areas. Therefore, the authors may not have a trustworthy history of these animals. Were these animals suffering from a history of disease, abortion, fever,………………..?

“All animals sampled were clinically healthy.” Did you examine all these animals clinically before slaughter? Did you confirm that their carcasses were accepted after visceral evaluation?

Why the author did not analyze the data in Table 1 statistically using chi-square?

The authors must clarify the importance of these data for the international community.

Reviewer #2: Partly acceptable

**Results**

-Does the analysis presented match the analysis plan?

-Are the results clearly and completely presented?

-Are the figures (Tables, Images) of sufficient quality for clarity?

Reviewer #1: Why the author did not analyze the data in Table 1 statistically using chi-square?

Reviewer #2: Partly acceptable

**Conclusions**

-Are the conclusions supported by the data presented?

-Are the limitations of analysis clearly described?

-Do the authors discuss how these data can be helpful to advance our understanding of the topic under study?

-Is public health relevance addressed?

Reviewer #1: (No Response)

Reviewer #2: Partly acceptable

**Editorial and Data Presentation Modifications?**

Reviewer #1: (No Response)

Reviewer #2: (No Response)

**Summary and General Comments**

Reviewer #1: Regarding the manuscript entitled “A comprehensive analysis of Coxiella burnetii infection (Q fever) and its risk factors in livestock at Addis Ababa, Adama, and Modjo abattoirs, and Pastoral areas of Oromia, Ethiopia: a seroepidemiological study”, the authors aimed to estimate the seroprevalence of C. burnetii infection and associated risk factors in cattle, sheep, goats, and camels, respectively from Addis Ababa, Adama, and Modjo abattoirs and pastoral areas of Oromia, Ethiopia.

English editing is needed for the whole manuscript. 

The study used a preliminary method for the diagnosis of Q fever. There is no molecular investigation. 

Samples were collected from abattoirs and three selected pastoral areas. Therefore, the authors may not have a trustworthy history of these animals. Were these animals suffering from a history of disease, abortion, fever,………………..?

“All animals sampled were clinically healthy.” Did you examine all these animals clinically before slaughter? Did you confirm that their carcasses were accepted after visceral evaluation?

Why the author did not analyze the data in Table 1 statistically using chi-square?

The authors must clarify the importance of these data for the international community.

Reviewer #2: This study is to investigate a seroepidemiology and risk factors of C. burnetii infection in four different species of animals in Ethiopia. This paper deserves to be published in Plos Journal. However, there are some critical concerns to make the reviewer hesitate to accept your article. 

Major comments

- Q fever mean C. burnetii infection in human, whereas coxiellosis is animal disease caused by C. burnetii. This study was only done in animals, so Q fever should be removed from study title. This study is only a cross-sectional seroprevalence study even though it was tough works in that a large-scale of serum sample was collected from four different areas and animals. The study title is better to be “A seroepidemiological analysis of coxiella burnetii infection and its risk factors among livestock in Ethiopia: A nationwide study 

- Journal guideline doesn’t have word count limitation, but it is recommended to present the study findings CONCISELY. However, this paper has about the number of 6,900 word, and the number of references is 74. Needless repetitions and redundant descriptions are seen all over the manuscript. If the authors will be able to concise the original draft and omit needless repetitions and redundant descriptions, this manuscript seems to be a well-written article. For this, it would be better to pay attention at least my following comments. To present study finding concisely and study suggestion clearly, all authors should review original draft carefully, and try to clarify study background, results and discussion. 

- In introduction, the description for study background is too long. Three or four paragraphs will be enough to construct three parts in introduction. It would be better to concise or reduce it up to two third in current amount of introduction. This manuscript is not a review article, but a research article. The number of references in the introduction section is too many. The author should cite only necessary references up to about 15. 

- In methods, description of study area is only supplementary findings, not directly involved in study methods. It would be better to move into supplementary material called supporting information including figure 1. 

- In line 332-337 of result section, this paragraph is not the study finding but methodologic description. It is clearly an example of needless repetitions. 

- In study findings, one of some critical concerns is about “animals infested with ticks” How did the authors define “animals infested with ticks”? In this study, Tick infestation is the greatest risk factor of seropositivity in animals. It can make readers misunderstand that C. burnetii infection in animals seems to be tick-borne disease. I can agree that tick can contribute to transmission among livestock and wildlife. However, given that human Q fever is primarily transmitted from livestock and transmission through tick is VERY unusual, this finding is likely to be biased. It should clarify the definition of tick infestation and consider re-analysis on the basis of clear study definition. 

- In the section subtitled “Analysis of risk factors of C. burnetii infection”, table 2 seems to be univariate analysis on association between seropositivity of C. burnetii infection and relevant variables. The authors should discuss statistical analysis with a statistician, and address statistician opinions in the revision letter. I recommend that OR of multivariate analysis along with OR of univariate analysis should be shown in table. 

- In line 383-384, there are two different but similar subheading. Please avoid the redundant description. 

- In line 383-413 subtitled “seroprevalence of C. burnetii at abattoris”, I think this section is not directly involved in study results, and it is much closer to background description of study area in the method section. Thus, it can move to supporting information. If the authors would like to remain these paragraphs in the result section, it should be removed the paragraphs of line 385-394 and line 409-413. Result section is for study findings, but not for methods and discussion. 

- In the discussion section, first paragraph of line 439-445 offer little that is associated with study results. This paragraph just addresses universal characteristics of C. burnetii infection. It can be removed from discussion section, and second paragraph of line 446-451 is much better to be first paragraph of discussion section.

- The sections of discussion and conclusion consist of 8 pages, and it is too long description. Needless repetitions and redundant descriptions are also seen in the discuss. Thus, It would be better to concise or reduce it withing 5 pages. Total number of references in this manuscript is 74, and it is too many. The author should cite only necessary references within 40. 

- The paragraph for recommendation should be removed from main manuscript. Previously said, the author should aware that this paper is only for research article, but not for narrative review.

PLOS authors have the option to publish the peer review history of their article (what does this mean?). If published, this will include your full peer review and any attached files.

Reviewer #1: Yes: Ayman A Swelum

Reviewer #2: No
---

## [Decision Letter · Decision Letter 1]

11 Mar 2024

Dear Dr. Dinka,

Thank you very much for submitting your manuscript "A seroepidemiological analysis of coxiella burnetii infection and its risk factors among livestock in Ethiopia: A nationwide study" for consideration at PLOS Neglected Tropical Diseases. As with all papers reviewed by the journal, your manuscript was reviewed by members of the editorial board and by several independent reviewers. In light of the reviews (below this email), we would like to invite the resubmission of a significantly-revised version that takes into account the reviewers' comments. 

We cannot make any decision about publication until we have seen the revised manuscript and your response to the reviewers' comments. Your revised manuscript is also likely to be sent to reviewers for further evaluation.

Sincerely,

Wen-Ping Guo

Academic Editor

Elsio Wunder Jr

Section Editor

Reviewer's Responses to Questions

**Key Review Criteria Required for Acceptance?**

**Methods**

-Are the objectives of the study clearly articulated with a clear testable hypothesis stated?

-Is the study design appropriate to address the stated objectives?

-Is the population clearly described and appropriate for the hypothesis being tested?

-Is the sample size sufficient to ensure adequate power to address the hypothesis being tested?

-Were correct statistical analysis used to support conclusions?

-Are there concerns about ethical or regulatory requirements being met?

Reviewer #3: Comments

- Comments:

- The title is captured as a national wide study yet sampling was only done in a few abattoirs and a pastoral area. A national wide study would mean that livestock population densities in the entire county was considered and weighted during sampling to capture regional differences, spatial distributions etc. Consider revising the title. 

- I also highly recommend English editing of the manuscript. 

- The authors should re-check the whole manuscript and remove redundant information in the abstract, author summary and many other sections. 

- Line 55 ‘‘prevalence rate’’ change to ‘prevalence estimate’ 

- Line 58-60: Please re-structure the sentence ‘‘The seroprevalence at abbatoirs and pastoral areas was found in the order of Addis Ababa (27.17%)>Adama (19.41%)>Modjo (19.13)>pastoral areas (12.1%), respectively’’. 

- Line 101-112: The information presented here is unreferenced.

- Lines 108- 123. Consider revising this paragraph as there are a number of repeated texts e.g., sampled species and age of sampled animals. 

- Line 124: Briefly, explain how the age of the study animals was estimated. 

- Lines 117. Consider deleting the word design as this is repeated in line 133. 

- Lines 122- 123. Consider moving the information about sample size to the section ‘sample size determination’ as this information is again repeated.

- Lines 127-128: Explain the selection criteria for the households. How were households selected?

- Line 139: It’s important to clarify that the expected prevalence of C. burnetii was assumed to be 50% for all targeted livestock species. Please also provide a justification for this. 

- Line 144: The included total number of animals (n = 4140) is approximately 2.7 times the estimated required sample size (n = 1536). Is there a justification why this was the case yet the sample size doesn’t seem to have been adjusted for possible clustering of exposures among animals in your sampling units, more so from the pastoral areas. Its unethical to over sample without any scientific justification. The variations in the number of animals sampled per species – could this affect the overall prevalence estimate?

- Line 149: The uppercase should not be in the middle of a subheading. 

- Lines 155- 160: Please remove repeated texts. Eg line ‘Each sample was given a code to identify the specific animal’ repeated in line 159-160. Pease do this in the entire document to avoid redundancy. 

- Lines 160-162: Clearly explain how serum was prepared. Was there centrifugation on site or a dedicated field laboratory? 

- Lines 156-157: This information should be presented at the end of this section. Please explain how this animal-level information was collected? 

- Line 169-170: Please specify on whether the ELISA kit used detected antibodies to phase 1 and/or 11 antigens of C. burnetii. 

- Line 181-184: Information about optical densities is repeated. Please revise accordingly. The information ‘‘the color development is directly proportional to the amount of C. burnetii’’ is unnecessary and should be deleted. 

- Lines 184-192: Please be concise and consider removing unnecessary details in this section. Cut-off values used to classify animals as either positive, negative or doubtful were based on what? This need to be clarified so that it does not appear that they were arbitrary cut off values. 

- Line 207: Remove the word infection. 

- Line 207: The paragraph starting with ‘these maps can provide ……..’ should be moved to the discussion or conclusion sections. 

- Lines 215- 216: ‘…….. estimate the effects of risk factors on the prevalence of C. burnetii infection’’. Consider re-writing this sentence. Did you mean to determine the risk factors associated with C. burnetii seropositivity?

- Lines 216- 217 ‘‘Descriptive statistics such as chi-squared test was to analyse the effect of individual factors on the prevalence of C. burnetii infection’’. This sentence should be corrected. Chi-squared test is a non-parametric test used to determine the independent associations between categorical factors and a dependent variable. 

- Lines 218: Were there interactions among the correlates in the multivariable model Please provide more details on the risk factor modelling methodology used. How did the presented multivariable model compare with a null intercept and parsimonious model without non-significant factors e.g., animal’s sex. How did you select the final model? 

- Lines 221-222: ‘‘The measure of association between prevalence and risk factors was reported as the odds ratio’’. This information is unnecessary and can be removed.

- Lines 222-224: The use of multilevel mixed-effects generalized linear models is not appropriate to your data given that a cluster sampling design was not used in this study or its not mentioned in the manuscript. What variable was used as a random effect in the modeling? More details on how data were fitted to these mixed effect models were conducted is not provided in the paper. Again, the results from these analyses were not presented. 

- Line 223: ‘A P value of < 0.05 and a 95% confidence interval were considered to be significant in this study’. Perhaps better indicate that variables with p values ≤ were considered statistically significant in the…...

**Results**

-Does the analysis presented match the analysis plan?

-Are the results clearly and completely presented?

-Are the figures (Tables, Images) of sufficient quality for clarity?

Reviewer #3: Comments

- Page 10. Table 1: Please create a column and provide the results of chi-squared test, the preliminary test done before multivariable logistic regression analyses. 

- Lines 229: Could you estimate the true prevalence estimates based on the sensitivity and specificity of the commercial ELISA kit used? 

- Line 248: use lower case for ‘p’ standing for p values in text. 

- Line 249: Remove the word ‘infection’. Lines 249-253: consider revising this section. For example, when you say … the odds of infection in camels was 3.22 (95% CI, 2.57-4.05%) times that of cattle….. You only detected antibodies to C. burnetii and no other diagnostic methods were used to determine if the animals had active infection. The presence of antibodies indicates prior exposures and not necessarily active infections. Please revise accordingly all the relevant sections. 

- Page 11: Please present the results to 2 dp to be consistent.

- Include a section on study limitation in the discussion section e.g., potential cross-reactions etc. 

- Line 413: Please reference the presented supplementary file in the manuscript.

**Conclusions**

-Are the conclusions supported by the data presented?

-Are the limitations of analysis clearly described?

-Do the authors discuss how these data can be helpful to advance our understanding of the topic under study?

-Is public health relevance addressed?

Reviewer #3: The author should avoid repeating prevalence estimates in this section.

**Editorial and Data Presentation Modifications?**

Reviewer #3: I would recommend a major revision of this manuscript as there are numerous repetitions in text. English editing of the paper is also needed.

**Summary and General Comments**

Reviewer #3: This study assessed the seroprevalence of C. burnetii in various livestock species in several abattoirs and in a pastoral area. The targeted pathogen is zoonotic and cause significant economic losses in livestock in endemic areas hence the need to understand its epidemiology. However, I suggest a major revision to improve the paper.

PLOS authors have the option to publish the peer review history of their article (what does this mean?). If published, this will include your full peer review and any attached files.

Reviewer #3: No
---

## [Editor Report · Decision Letter 2]

11 Jun 2024

Dear Dr. Dinka,

We are pleased to inform you that your manuscript 'A sero-epidemiological analysis of Coxiella burnetii infection and its risk factors in livestock from Addis Ababa, Adama, and Modjo abattoirs and pastoral areas of Oromia, Ethiopia' has been provisionally accepted for publication in PLOS Neglected Tropical Diseases.

Best regards,

Wen-Ping Guo

Academic Editor

Elsio Wunder Jr

Section Editor

---

## [Editor Report · Acceptance letter]

28 Jun 2024

Dear Dr. Dinka,

We are delighted to inform you that your manuscript, "A sero-epidemiological analysis of Coxiella burnetii infection and its risk factors in livestock from Addis Ababa, Adama, and Modjo abattoirs and pastoral areas of Oromia, Ethiopia," has been formally accepted for publication in PLOS Neglected Tropical Diseases.

Best regards,

Shaden Kamhawi

co-Editor-in-Chief

Paul Brindley

co-Editor-in-Chief
